# Transparency of COVID-19-related research: A meta-research study

Ahmad Sofi-Mahmudi [1,2,3]*, Eero Raittio[4,5], Sergio E. Uribe[6,7,8]

**1** National Pain Centre, Department of Anesthesia, McMaster University, Hamilton, Ontario, Canada, **2** Department of Health Research Methods, Evidence and Impact, McMaster University, Hamilton, Ontario, Canada, **3** Seqiz Health Network, Kurdistan University of Medical Sciences, Seqiz, Kurdistan, **4** Institute of Dentistry, University of Eastern Finland, Kuopio, Finland, **5** Department of Dentistry and Oral Health, Aarhus University, Aarhus, Denmark, **6** Department of Conservative Dentistry and Oral Health, Riga Stradins University, Riga, Latvia, **7** School of Dentistry, Universidad Austral de Chile, Valdivia, Chile, **8** Baltic Biomaterials Centre of Excellence, Headquarters at Riga Technical University, Riga, Latvia

\* sofima@mcmaster.ca, a.sofimahmudi@gmail.com.

## Abstract

### Background

We aimed to assess the adherence to five transparency practices (data availability, code availability, protocol registration and conflicts of interest (COI), and funding disclosures) from open access Coronavirus disease 2019 (COVID-19) related articles.

### Methods

We searched and exported all open access COVID-19-related articles from PubMed-indexed journals in the Europe PubMed Central database published from January 2020 to June 9, 2022. With a validated and automated tool, we detected transparent practices of three paper types: research articles, randomized controlled trials (RCTs), and reviews. Basic journal- and article-related information were retrieved from the database. We used R for the descriptive analyses.

### Results

The total number of articles was 258,678, of which we were able to retrieve full texts of 186,157 (72%) articles from the database Over half of the papers (55.7%, n = 103,732) were research articles, 10.9% (n = 20,229) were review articles, and less than one percent (n = 1,202) were RCTs. Approximately nine-tenths of articles (in all three paper types) had a statement to disclose COI. Funding disclosure (83.9%, confidence interval (CI): 81.7–85.8 95%) and protocol registration (53.5%, 95% CI: 50.7–56.3) were more frequent in RCTs than in reviews or research articles. Reviews shared data (2.5%, 95% CI: 2.3–2.8) and code (0.4%, 95% CI: 0.4–0.5) less frequently than RCTs or research articles. Articles published in 2022 had the highest adherence to all five transparency practices. Most of the reviews (62%) and research articles (58%) adhered to two transparency practices, whereas almost half of the RCTs (47%) adhered to three practices. There were journal- and publisher-

**Data Availability Statement:** All the datasets and codes generated and analysed during the current study are available on our OSF (osf.io/x3kb6) and GitHub (github.com/choxos/covid-transparency) repositories.

**Funding:** The computational analyses were performed on servers provided by UEF Bioinformatics Center, University of Eastern Finland, Finland. Uribe was supported by European Union's Horizon 2020 grant 857287 for the Baltic Biomaterials Centre of Excellence, Headquarters at Riga Technical University, Riga, Latvia and the Uzņēmuma MikroTik līgumam Nr. UL8, 2021 RSU (toward implementing the RSU data repository and the FAIR data management principles). Raittio was supported by the Finnish Dental Society Apollonia and the Aarhus University Research Foundation (#AUFF-E 2019-7-3).

**Competing interests:** The authors have declared that no competing interests exist.

**Abbreviations:** COI, conflict of interest; CORD-19, COVID-19 Open Research Dataset; COVID-19, coronavirus disease 2019; EPMC, Europe PubMed Central; JIF, journal impact factor; L·OVE, Living OVerview of Evidence; OECD, Organisation for Economic Co-operation and Development; OSF, Open Science Framework; PMID, PubMed ID; RCT, randomized controlled trial; SJR, SCImago Journal Rank.

related differences in all five practices, and articles that did not adhere to transparency practices were more likely published in lowest impact journals and were less likely cited.

## Conclusion

While most articles were freely available and had a COI disclosure, adherence to other transparent practices was far from acceptable. A much stronger commitment to open science practices, particularly to protocol registration, data and code sharing, is needed from all stakeholders.

## Background

Access to research publications, their underlying data and methods that enable the reuse and reproduction of the research are core features of open science. For instance, research data that are findable, accessible, interoperable, and reusable (FAIR principles) are expected to facilitate knowledge discovery, promoting collaboration across different research communities and advancing scientific research by reducing barriers to data sharing [1, 2]. In this context, practices like protocol registration and disclosing conflict of interest (COI) funding help ensure scientific research integrity. Without research integrity, the credibility and reliability of scientific findings and underlying data and methods may be compromised.

However, during the recent coronavirus disease 2019 (COVID-19) pandemic, some public members in developed countries have cited a lack of transparency in scientific studies used to justify public health measures [3]. Thus, international organizations like the Organisation for Economic Co-operation and Development (OECD) have stressed the importance of transparent communication with citizens to support public health measures and counter misinformation [4]. To facilitate rapid and collaborative scientific research, over 100 organizations—including universities, publishers, funders, and journals—signed a statement in January 2020 supporting unrestricted access to research data, tools, and other information related to COVID-19 [5].

The COVID-19 pandemic has also produced an enormous volume of scientific publications across various fields of study, leading to the development of vaccines and the evaluation of community interventions. While transparent scientific practices such as data and code sharing have helped combat the pandemic globally (e.g., by allowing information absorption via the established common data repositories and international and interdisciplinary collaborations) [6–8], the lack of transparency in some key developments and the rampant dis/misinformation (fake news, conspiracy theories) have contributed to public mistrust of research and public health measures [3]. This indicates a lack of awareness regarding the extent of transparency practices in COVID-19-related medical literature. As an example, it is unclear whether open science initiatives related to the COVID-19 pandemic [6, 7] to advance data sharing have led to a higher number of articles sharing data in COVID-19 research compared to what has been seen in studies in general in biomedical literature [9].

We aimed to programmatically assess the adherence to transparent scientific practices (data sharing, code sharing, conflict of interest (COI), disclosure, funding disclosure, and protocol registration) from open access full text COVID-19-related articles published in PubMed-indexed journals from the Europe PubMed Central (EPMC) database.

## Methods

The protocol of this descriptive study was published beforehand on the Open Science Framework (OSF) website (https://osf.io/5kx2n). All code and data related to the study were shared via its OSF repository (https://osf.io/x3kb6) and GitHub (https://github.com/choxos/covid-transparency) at the time of submission of the manuscript. Deviations from the protocol are available in S1 Text.

### Data sources and study selection

First, we searched for all open access PubMed-indexed records available in the EPMC database from 1/1/2020 to 9/6/2022. This database included all the records available through PubMed and PubMed Central and also enabled us to retrieve the record automatically, which is not available through the PubMed website. Then, we used the LitCovid database (https://ncbi.nlm.nih.gov/research/coronavirus) to detect COVID-19-related papers. LitCovid, sponsored by the National Library of Medicine, is a curated literature hub to track up-to-date COVID-19-related scientific information in PubMed. LitCovid is updated daily with newly identified relevant articles organized into curated categories. It uses machine learning and deep-learning algorithms [10, 11]. We merged both datasets using the PubMed IDs (PMIDs) of the records.

We then downloaded the full text of all identified open access COVID-19-related available records in XML format using the *metareadr* package [12] from the EPMC database.

We used the EPMC publication type variable to detect research articles and reviews. We used the "research articles" filter of EPMC (in publication type) to identify papers that have used and analyzed data (of any kind) and to exclude opinions, commentaries, and letters. We combined two publication types to identify reviews: "review" and "systematic-review". As the EPMC's publication type is not comprehensive for randomized controlled trials (RCTs) and many of them are not labelled correctly, to detect RCTs, we used the Living OVerview of Evidence (L·OVE) platform (iloveevidence.com). L·OVE, powered by Epistemonikos Foundation, maps and organizes all of the best evidence in various medical and health sciences fields. It has a database for COVID-19-related papers in some categories, including RCTs. This database has been seen to be very comprehensive [13]. Its RCTs database includes both protocols and papers with the results. As we want to focus solely on RCTs that have been done, we applied two filters: "RCT" and "Reporting data," on the L·OVE website and then downloaded the dataset with these characteristics. We used PMIDs of COVID-19-related RCTs provided in this downloaded dataset from the L·OVE website to detect RCTs in our main dataset of all open access COVID-19-related papers.

### Data extraction and synthesis

We assessed articles' adherence to five transparent practices:

1. data sharing,

2. code sharing,

3. COI disclosures,

4. funding disclosures,

5. protocol registration.

We used a validated and automated tool developed by Serghiou et al. [9] suitable to identify these five transparent practices from articles in XML format from the EPMC database. Briefly, this tool uses some keywords to identify adherence to each indicator using regular expressions.

For example, it searches for phrases commonly associated with a COI disclosure (e.g., "conflicts of interest," "competing interests," "Nothing to disclose.," etc.) in the body or titles of the sections of the text file of an article. For COI and funding disclosure, this tool only determines any mentions of disclosure, whether there was anything to disclose or not. For instance, a paper with a phrase such as "Nothing to disclose" was considered transparent, just like when there was something to disclose. For data and code sharing detection, it both detects shared as a supplement or shared in a general (e.g., figshare, OSF, GitHub, etc.) or field-specific repository (e.g., dbSNP, ProteomeXchange, GenomeRNAi, etc.) as adherence to transparency to data/code sharing. However, those articles that indicated "data available under request" would be classified as no data available since it is unlikely to obtain these data [14]. Overall, all the mentions of the indicators were classified as affirmative when the article explicitly contained them.

For validation of the identification of transparency practices in the sample articles, we manually checked the five transparency indicators from 100 random articles in the sample with methods described by Serghiou et al. [9] (available on OSF as covid_transparency_random-sample.csv or in S1 Appendix). Sensitivities and specificities of the tool for detecting each open science practice as provided by the developers were as follows:

- Data sharing [sensitivity 0.76 (95% confidence interval (CI): 0.61–0.94), specificity 0.99 (95% CI: 0.98–1.00)];

- Code sharing [sensitivity 0.59 (95% CI: 0.34–0.94), specificity 1.00 (95% CI: 1.00–1.00)];

- COI disclosures [sensitivity 0.99 (95% CI: 0.99–0.99), specificity 1.00 (95% CI: 0.99–1.00)];

- Funding disclosures [sensitivity 1.00 (95% CI: 0.99–1.00), specificity 0.98 (95% CI: 0.96–1.00)]; and,

- Protocol registration [sensitivity 0.96 (95% CI: 0.92–0.99), specificity 1.00 (95% CI: 1.00–ß1.00)].

Basic journal- and article-related information (publisher, publication year, citations to article and journal name) were retrieved from the EPMC database. The impact index of the journals (JIFs) was obtained from the Journal Citation Reports 2021 (https://jcr.clarivate.com) and the SCImago Journal Rank 2020 (SJR) and H-index indicators from the SCImago website (https://scimagojr.com). We also calculated the proportion of articles available as open access full texts via the EPMC from the total number of COVID-19-related articles in the database.

## Data analysis

We used R v4.1.2 [15] for searches, data handling, analysis and reporting. The searches and data export from the EPMC were conducted with the *europepmc* package [16]. Indicators of transparency practices from the available full texts were extracted with the *rtransparent* package [17]. Trends over time in transparency practices were reported using descriptive tabulations and visualizations using the *ggplot2* package [18]. We used the sensitivity and specificity of the *rtransparent* package [9] to generate 95% CIs for our prevalence estimates of the transparency practices with the *epiR* package [19]. We determined the level of adherence to transparency practices in articles, ranging from 0 to 5 practices. Generalized linear models with logit link were used to analyze transparency indicators by JIF, SJR, and H-index or received citations in research articles. These models were adjusted for the month-year of publication and whether the article was RCT. The Fisher's exact test with Monte-Carlo simulated p-values for differences in transparency practices by journals and publishers were performed. We

reported the interquartile range (the third quartile (Q3)–the first quartile (Q1)) and median (instead of mean and standard deviation) when the data were skewed.

## Results

### General characteristics

As of June 9, 2022, there were 258,678 COVID-19-related articles, including open access and non-open access publications. Of those, full texts of 186,279 (72.0%) articles were accessible via the EPMC. However, 122 (0.1%) of these articles were not downloadable because of technical issues and were excluded from our analyses. Consequently, the sample included 186,157 full text articles. Fig 1 shows the Venn diagram of the study.

Out of all included articles, 65,316 (35.1%) were published in 2020, 89,222 (47.9%) in 2021, and 31,619 (17.0%) in 2022 by June 9. More than half of the papers (n = 103,732, 55.7%) were research articles, followed by letters (n = 22,449, 12.1%), and review articles (n = 20,229, 10.9%), and less than one percent (n = 1,202) were RCTs (Fig 2). The 186,157 articles were from 6,582 different journals, with the top three being the International Journal of Environmental Research and Public Health (n = 4,289, 2.3%), PLoS ONE (n = 3,545, 1.9%), and Cureus (n = 2,065, 1.1%). According to our random sample of research articles, most of the papers were observational studies (n = 53).

### Transparency practices

We found that 91,776 (88.5%) of the research articles had a statement to disclose COI, and the prevalence was higher in RCTs (92.6%) and reviews (91.9%). Funding disclosures were detected in 76,481 (73.7%) research articles, more frequently in RCTs (83.9%) and less

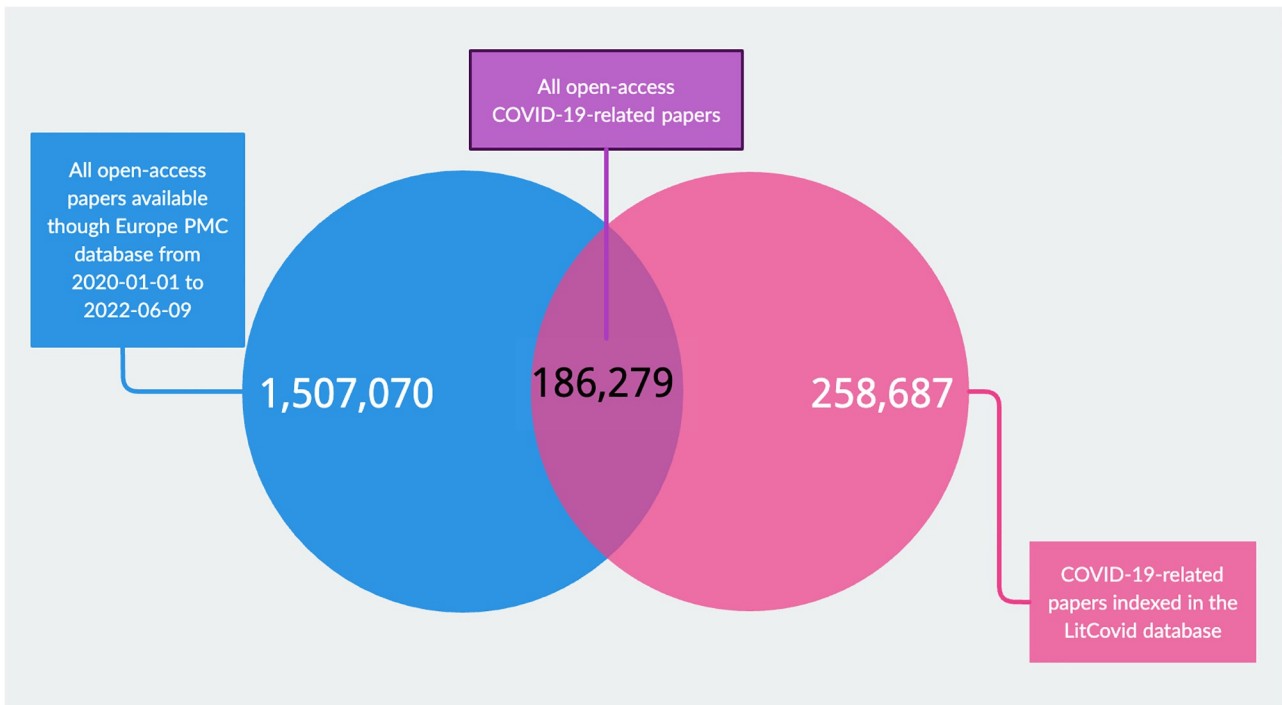

**Fig 1. The Venn diagram of the study.**

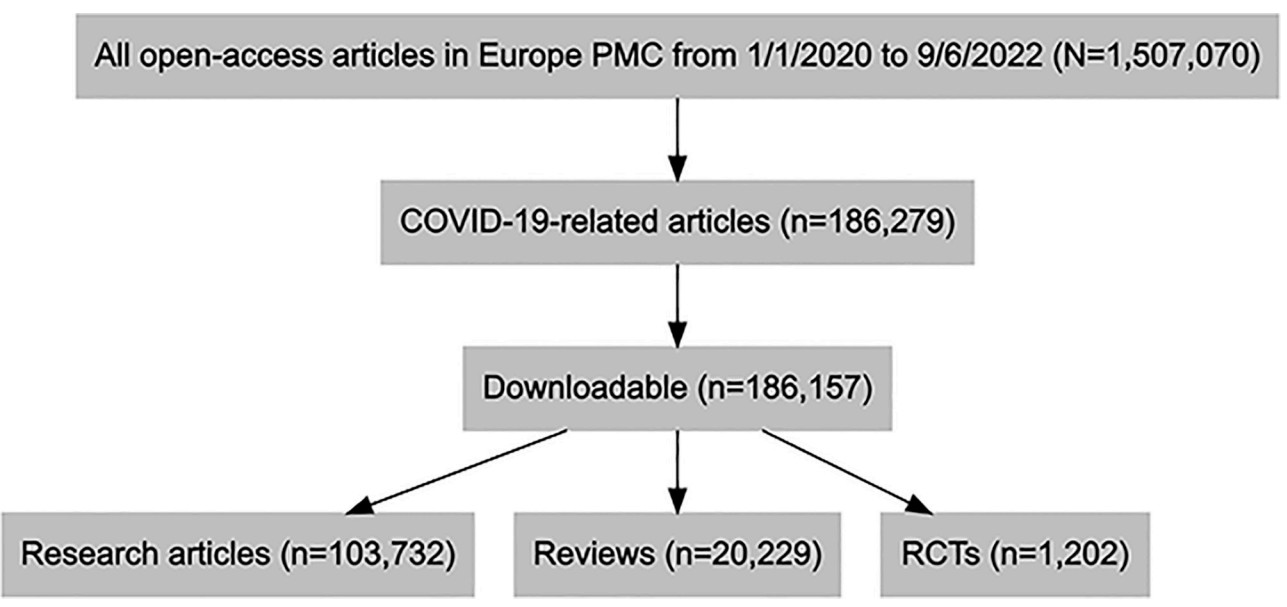

**Fig 2. The flow diagram of the study.**

frequently in reviews (69.0%). One in 25 research articles was registered beforehand (n = 4,362), and the proportion of registered articles was manifold higher in RCTs (53.5%) than in reviews (4.9%) or research articles (4.2%). About one in ten research articles (n = 11,599, 11.2%) and RCTs (11.7%) shared data, whereas this proportion was four times lower in reviews (2.5%). One in 25 research articles shared code (n = 4,192, 4%). The proportion was lower in RCTs (0.7%) and reviews (0.4%) than in research articles. More information is illustrated in the left-hand bar charts in Fig 3 and Table 1.

Research articles and reviews published in 2022 adhered the most to all five transparency practices (Table 1, Fig 3-right). We did not find a clear time trend for RCTs (Fig 3B-right). Adherence to the five transparency practices was identified in <0.1% of articles. Adherence to three practices was detected in 47% of RCTs and two practices in 62% of reviews (Table 2).

There were journal and publisher-related differences in all five practices (Tables 3 and 4). All the papers from MDPI and PLoS had COI and funding disclosures. Whereas COI and funding disclosures were available for almost all the RCTs published in the New England Journal of Medicine and The Lancet, they rarely shared their data (0% and 2.9%, respectively). The highest percentage of data sharing was found in PLoS One research articles (54.8%) and RCTs (70.0%). RCTs published in The Lancet and The Lancet Respiratory Medicine had the highest adherence to protocol registration, with 94.1% and 100%, respectively. Code sharing was a rare practice, with Scientific Reports (14.2%) and PLoS (11.1%) being the top journal and the top publisher for adherence to this practice. None of the journals with the highest number of RCTs in the sample shared their codes.

Research articles in the lowest JIF or SJR quintiles were least likely to adhere to the transparency practices. Regarding protocol registration, the differences were the smallest between the JIF or SJR quintiles (Fig 4A and 4C). The more citations articles received, the more likely the article adhered to the transparency practices (Fig 4B). Detailed information for RCTs and reviews is available in S1–S4 Tables.

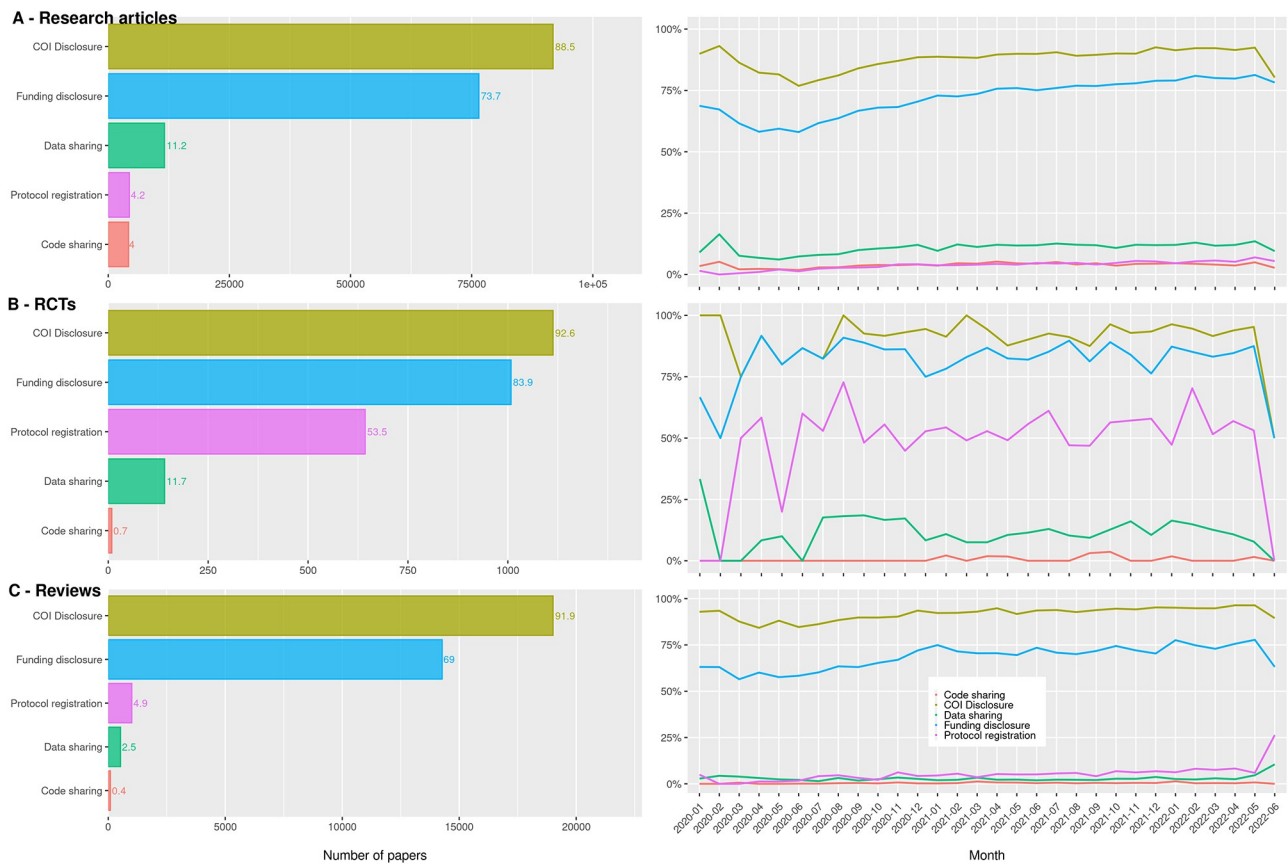

**Fig 3. The proportion of adherence to transparency practices and monthly trends for each indicator.** Left-side bar charts: number of articles adhering to transparency practices and proportion from the total. Right-side line graphs: proportion of articles adhering to transparency practices over time. A) All research articles; B) Randomized controlled trials (RCTs); C) Reviews.

**Table 1. Transparency practices by each year (%).**

|  | Year | COI disclosure | Funding disclosure | Protocol registration | Data sharing | Code sharing |
|---|---|---|---|---|---|---|
| Research articles (N = 103,732) | 2020, N = 27,890 | 83.5 (83.0–83.9) | 64.7 (64.2–65.3) | 2.8 (2.6–3.0) | 9.3 (9.0–9.6) | 3.2 (3.0–3.4) |
|  | 2021, N = 54,557 | 89.7 (89.5–90.0) | 75.8 (75.5–76.2) | 4.5 (4.3–4.6) | 11.7 (11.4–12.0) | 4.4 (4.2–4.6) |
|  | 2022, N = 21,285 | 91.8 (91.5–92.2) | 80.1 (79.6–80.7) | 5.5 (5.2–5.8) | 12.4 (11.9–12.8) | 4.3 (4.0–4.5) |
|  | Overall | 88.5 (88.3–88.7) | 73.7 (73.5–74.0) | 4.2 (4.1–4.3) | 11.2 (11.0–11.4) | 4.0 (3.9–4.2) |
|  | *P-value* | <0.001 | <0.001 | <0.001 | <0.001 | <0.001 |
| RCTs (N = 1,202) | 2020, N = 168 | 90.5 (85.1–94.1) | 85.1 (79.0–89.7) | 49.4 (41.9–56.9) | 14.3 (9.8–20.4) | 0 (0–2.2) |
|  | 2021, N = 679 | 92.5 (90.3–94.2) | 82.9 (79.9–85.6) | 53.3 (49.6–57.0) | 10.8 (8.6–13.3) | 1.0 (0.5–2.1) |
|  | 2022, N = 355 | 93.8 (90.8–95.9) | 85.1 (81.0–88.4) | 55.8 (50.6–60.9) | 12.4 (9.4–16.2) | 0.6 (0.2–2.0) |
|  | Overall | 92.6 (91.0–93.9) | 83.9 (81.7–85.8) | 53.5 (50.7–56.3) | 11.7 (10.0–13.7) | 0.7 (0.4–1.4) |
|  | *P-value* | 0.393 | 0.598 | 0.391 | 0.399 | 0.340 |
| Reviews (N = 20,682) | 2020, N = 7,753 | 88.6 (87.8–89.2) | 62.9 (61.9–64.0) | 3.2 (2.9–3.7) | 2.5 (2.2–2.9) | 0.3 (0.2–0.4) |
|  | 2021, N = 9,784 | 93.5 (93.0–93.9) | 71.8 (70.9–72.6) | 5.3 (4.9–5.8) | 2.4 (2.2–2.8) | 0.5 (0.4–0.7) |
|  | 2022, N = 3,145 | 95.4 (94.6–96.0) | 75.4 (73.9–76.9) | 7.5 (6.6–8.4) | 2.9 (2.4–3.6) | 0.6 (0.4–0.9) |
|  | Overall | 91.9 (91.5–92.3) | 69.0 (68.4–69.6) | 4.9 (4.6–5.2) | 2.5 (2.3–2.8) | 0.4 (0.4–0.5) |
|  | *P-value* | <0.001 | <0.001 | <0.001 | 0.316 | 0.008 |

Numbers in percentage (%). Confidence intervals (CIs) in square brackets. P-values from Pearson's Chi-squared test. COI: conflict of interest.

**Table 2. Adherence to transparency practices by the number of total practices.**

| Study type | None | 1 | 2 | 3 | 4 | All |
|---|---|---|---|---|---|---|
| Research articles (N = 103,732) | 7,560 (7.3%) | 21,617 (20.8%) | 59,740 (57.6%) | 11,997 (11.6%) | 2,768 (2.7%) | 50 (<0.1%) |
| RCTs (N = 1,202) | 43 (3.6%) | 91 (7.6%) | 443 (36.9%) | 564 (46.9%) | 60 (5.0%) | 1 (<0.1%) |
| Reviews (N = 20,682) | 1,106 (5.3%) | 5,535 (26.8%) | 12,826 (62.0%) | 1,145 (5.5%) | 68 (0.3%) | 2 (<0.1%) |

## Validation sample

Of 100 random articles for validation, 20 discrepancies (out of 500 (five indicators per article), 4%) between the automatic tool and manual checking were found: 9% for open data [sensitivity 0.95 (95% CI: 0.89–0.99), specificity 0.62 (95% CI: 0.32–0.86)], 2% for open code [sensitivity 0.99 (95% CI: 0.94–1.00), specificity 0.67 (95% CI: 0.09–0.99)], 1% for COI disclosure [sensitivity 1.00 (95% CI: 0.72–1.00), specificity 0.99 (95% CI: 0.94–1.00)], 6% for funding disclosure [sensitivity 0.86 (95% CI: 0.68–0.96), specificity 0.97 (95% CI: 0.90–1.00)], and 2% for registration [sensitivity 0.99 (95% CI: 0.94–1.00), specificity 0.50 (95% CI: 0.01–0.99)] (S1 Appendix).

## Discussion

Our study showed that adherence to transparent practices increased in COVID-19-related medical literature from 2020 to 2022. Adherence to reporting COI disclosure was high throughout the study period. In addition, most articles had funding disclosure. Data sharing, code sharing and protocol registration were rare but improved little over the study period. Higher adherence to COI disclosure and protocol registration was seen in randomized trials and reviews than in other research articles. Most research papers and reviews adhered to two or fewer indicators, whereas most RCTs adhered to three or fewer indicators. Only 53 papers adhered to all indicators overall. Journal- and publisher-related differences in transparency practices were clear. While some journals and publishers were completely transparent regarding COI and funding disclosures, they performed poorly for the other three indicators. In addition, journals with the lowest JIFs and SJRs seemed to publish less transparent articles. Articles which received more citations were more likely to have adhered to transparency practices than articles with fewer citations.

In general, adherence to transparent practices, except for COI disclosure, was at a similar level in COVID-19-related literature than in other biomedical literature analyzed with the same methods (see Fig 3 in [9] and also [20, 21]). This is surprising, particularly when considering worldwide, remarkable and noble initiatives to enhance open science to tackle the pandemic. As early as January 2020, over 100 organizations, including journals, publishers, funders, universities, and other institutions, signed a statement to ensure free access to research data, tools, and other information related to COVID-19 [5, 7]. Later, other initiatives to support the goal emerged, for instance, the COVID-19 Open Research Dataset (CORD-19) [22], a free resource of over 280,000 articles about the COVID-19 virus. However, it is possible that the algorithms used here did not efficiently detect all the different ways of sharing data and material that emerged after the pandemic because the algorithms were validated before the pandemic [9]. Furthermore, any initiative and movement need time to be effective. Another reason could be that publishers and journals should implement these changes.

Lacking protocol registration (for RCTs and reviews in particular), code and data sharing, and COI disclosure got attention during the pandemic [6, 23, 24]. However, we are unaware of the investigation of transparency in these aspects in COVID-19-related research on this scale in medical research. Comparing our study to other studies on the transparency of COVID-

**Table 3. Transparency practices in the five most common journals in the sample that published research articles, randomized controlled trials (RCTs), and reviews (%).**

| | Journal | COI disclosure | Funding disclosure | Protocol registration | Data sharing | Code sharing |
|---|---|---|---|---|---|---|
| Journals with highest number of research articles in the sample (N = 103,732) | Int J Environ Res Public Health, N = 3,671 | 100 (99.9–100) | 100 (99.9–100) | 2.6 (2.1–3.2) | 6.0 (5.2–6.8) | 1.6 (1.3–2.1) |
| | PLoS One, N = 3,492 | 100 (99.9–100) | 100 (99.9–100) | 5.6 (4.8–6.4) | 54.8 (53.1–56.4) | 8.6 (7.8–9.6) |
| | Sci Rep, N = 1,966 | 99.5 (99.1–99.8) | 79.7 (77.9–81.4) | 4.6 (3.8–5.6) | 19.8 (18.1–21.7) | 14.2 (12.8–15.9) |
| | Front Psychol, N = 1,366 | 100 (99.7–100) | 59.7 (57.1–62.3) | 1.8 (1.2–2.6) | 5.3 (4.3–6.7) | 1.0 (0.6–1.7) |
| | Front Public Health, N = 1,042 | 100 (99.6–100) | 66.9 (64.0–69.7) | 2.2 (1.5–3.3) | 3.6 (2.7–5.0) | 2.1 (1.4–3.2) |
| | Other, N = 92,195 | 87.0 (86.8–87.3) | 71.8 (71.6–72.1) | 4.3 (4.1–4.4) | 9.7 (9.5–9.9) | 3.8 (3.7–3.9) |
| | *P-value* | <0.001 | <0.001 | <0.001 | <0.001 | <0.001 |
| Journals with the highest number of RCTs in the sample (N = 1,202) | N Engl J Med, N = 56 | 100 (93.6–100) | 100 (93.6–100) | 50.0 (37.3–62.7) | 0.0 (0.0–6.4) | 0.0 (0.0–6.4) |
| | Lancet, N = 34 | 97.1 (85.1–99.8) | 94.1 (80.9–98.4) | 94.1 (80.9–98.4) | 2.9 (0.2–14.9) | 0.0 (0.0–10.2) |
| | Int J Environ Res Public Health, N = 33 | 100 (89.6–100) | 100 (89.6–100) | 39.4 (24.7–56.3) | 15.2 (6.7–30.9) | 0.0 (0.0–10.4) |
| | EClinicalMedicine, N = 33 | 93.9 (80.4–98.3) | 90.9 (76.4–96.9) | 75.8 (59.0–87.2) | 18.2 (8.6–34.4) | 0.0 (0.0–10.4) |
| | PLoS One, N = 30 | 100 (88.6–100) | 100 (88.6–100) | 36.7 (21.9–54.5) | 70.0 (52.1–83.3) | 0.0 (0.0–11.4) |
| | Lancet Respir Med, N = 30 | 100 (88.6–100) | 96.7 (83.3–99.8) | 100 (88.6–100) | 0.0 (0.0–11.4) | 0.0 (0.0–11.4) |
| | Other, N = 986 | 91.3 (89.4–92.9) | 80.9 (78.4–83.3) | 51.1 (48.0–54.2) | 11.0 (9.2–13.1) | 0.9 (0.5–1.7) |
| | *P-value* | 0.011 | <0.001 | <0.001 | <0.001 | >0.9 |
| Journals with the highest number of reviews in the sample (N = 20,682) | Int J Environ Res Public Health, N = 384 | 100 (99.0–100) | 100 (99.0–100) | 11.2 (8.4–14.7) | 2.6 (1.4–4.7) | 0.5 (0.1–1.9) |
| | Front Immunol, N = 347 | 100 (98.9–100) | 77.2 (72.5–81.3) | 1.2 (0.4–2.9) | 0.6 (0.2–2.1) | 0.3 (0.0–1.6) |
| | Int J Mol Sci, N = 341 | 100 (98.9–100) | 99.1 (97.4–99.7) | 0.0 (0.0–1.1) | 3.2 (1.8–5.7) | 0.3 (0.0–1.6) |
| | J Clin Med, N = 231 | 100 (98.4–100) | 99.1 (96.9–99.8) | 8.2 (5.3–12.5) | 3.5 (1.8–6.7) | 0.4 (0.0–2.4) |
| | Vaccines (Basel), N = 211 | 100 (98.2–100) | 100 (98.2–100) | 7.1 (4.4–11.4) | 2.8 (1.3–6.1) | 0.5 (0.0–2.6) |
| | Viruses, N = 211 | 100 (98.2–100) | 100 (98.2–100) | 0.9 (0.3–3.4) | 2.8 (1.3–6.1) | 0.5 (0.0–2.6) |
| | Other, N = 18,957 | 91.3 (90.9–91.7) | 67.0 (66.3–67.7) | 4.8 (4.5–5.1) | 2.5 (2.3–2.8) | 0.4 (0.4–0.5) |
| | *P-value* | <0.001 | <0.001 | <0.001 | 0.3 | >0.9 |

Numbers in percentage (%). Confidence intervals (CIs) in square brackets. P-values from Pearson's Chi-squared test; Fisher's Exact Test for Count Data with simulated p-value (based on 2000 replicates. COI: conflict of interest.

19-related research is difficult due to the different methodologies and the scale of our research. Nevertheless, it seems that data sharing was a little less common in our sample than the proportion detected at the beginning of the pandemic measured by Lucas-Dominguez et al. in PubMed Central [25]. In addition, the proportion of studies that shared data in our study

**Table 4. Transparency practices in the six most common publishers of journals in the sample.**

| | Publisher | COI disclosure | Funding disclosure | Protocol registration | Data sharing | Code sharing |
|---|---|---|---|---|---|---|
| Publishers with highest number of research articles in the sample (N = 88,460*) | MDPI, N = 6,498 | 100 (99.9–100) | 100 (99.9–100) | 3.0 (2.6–3.4) | 9.5 (8.8–10.2) | 2.9 (2.6–3.4) |
| | Frontiers, N = 4,489 | 100 (99.9–100) | 69.3 (67.9–70.6) | 3.0 (2.6–3.6) | 6.3 (5.7–7.1) | 1.8 (1.5–2.2) |
| | Elsevier, N = 4,227 | 91.1 (90.2–91.9) | 80.5 (79.3–81.6) | 2.4 (1.9–2.9) | 11.6 (10.7–12.6) | 3.5 (3.0–4.1) |
| | BioMed Central, N = 4,181 | 100 (99.9–100) | 99.4 (99.1–99.6) | 13.4 (12.4–14.5) | 9.4 (8.6–10.3) | 5.0 (4.4–5.7) |
| | Public Library of Science, N = 3,935 | 100 (99.9–100) | 100 (99.9–100) | 5.7 (5.0–6.4) | 55.3 (53.8–56.9) | 11.1 (10.1–12.1) |
| | Other, N = 65,130 | 85.6 (85.4–85.8) | 68.9 (68.6–69.2) | 3.9 (3.8–4.1) | 9.5 (9.3–9.7) | 3.9 (3.8–4.0) |
| | *P-value* | <0.001 | <0.001 | <0.001 | <0.001 | <0.001 |
| Publishers with the highest number of RCTs in the sample (N = 1,051*) | Elsevier Ltd., N = 86 | 95.3 (88.6–98.2) | 94.2 (87.1–97.5) | 80.2 (70.6–87.3) | 7.0 (3.2–14.4) | 0.0 (0.0–4.3) |
| | Frontiers, N = 60 | 100 (94.0–100) | 88.3 (77.8–94.2) | 25.0 (15.8–37.2) | 3.3 (0.9–11.4) | 1.7 (0.1–8.9) |
| | Lancet, N = 59 | 96.6 (88.5–99.1) | 91.5 (81.6–96.3) | 86.4 (75.5–93.0) | 10.2 (4.7–20.5) | 0.0 (0.0–6.1) |
| | BioMed Central, N = 57 | 98.2 (90.7–99.9) | 98.2 (90.7–99.9) | 71.9 (59.2–81.9) | 10.5 (4.9–21.1) | 0.0 (0.0–6.3) |
| | Massachussetts Medical Society, N = 56 | 100 (93.6–100) | 100 (93.6–100) | 50.0 (37.3–62.7) | 0.0 (0.0–6.4) | 0.0 (0.0–6.4) |
| | MDPI, N = 56 | 100 (93.6–100) | 100 (93.6–100) | 46.4 (34.0–59.3) | 10.7 (5.0–21.5) | 0.0 (0.0–6.4) |
| | Other, N = 677 | 90.1 (87.9–91.9) | 78.7 (75.8–81.4) | 49.9 (46.5–53.3) | 13.9 (11.7–16.4) | 1.0 (0.5–1.9) |
| | *P-value* | <0.001 | <0.001 | <0.001 | 0.007 | >0.9 |
| Publishers with the highest number of reviews in the sample (N = 16,943*) | MDPI, N = 1,797 | 100 (99.8–100) | 99.8 (99.4–99.9) | 3.3 (2.6–4.3) | 2.8 (2.2–3.7) | 0.4 (0.2–0.9) |
| | Frontiers, N = 1,179 | 100 (99.7–100) | 67.1 (64.4–69.7) | 5.9 (4.7–7.4) | 1.6 (1.0–2.5) | 0.4 (0.2–1.0) |
| | Elsevier BV, N = 808 | 91.7 (89.6–93.4) | 74.1 (71.0–77.0) | 4.6 (3.3–6.2) | 2.6 (1.7–3.9) | 0.2 (0.1–0.9) |
| | Elsevier, N = 800 | 90.2 (88.0–92.1) | 74.5 (71.4–77.4) | 3.9 (2.7–5.4) | 4.8 (3.5–6.5) | 0.5 (0.2–1.3) |
| | Elsevier Ltd., N = 624 | 82.1 (78.8–84.9) | 68.6 (64.8–72.1) | 2.2 (1.3–3.7) | 4.5 (3.1–6.4) | 0.6 (0.2–1.6) |
| | Other, N = 11,735 | 90.8 (90.4–91.3) | 65.0 (64.3–65.8) | 5.1 (4.8–5.5) | 2.4 (2.1–2.6) | 0.4 (0.3–0.6) |
| | *P-value* | <0.001 | <0.001 | <0.001 | <0.001 | >0.9 |

Numbers in percentage (%). Confidence intervals (CIs) in square brackets. P-values from Pearson's Chi-squared test; Fisher's Exact Test for Count Data with simulated p-value (based on 2000 replicates).

*Not all articles were published in journals with available publisher information. COI: conflict of interest.

sample was lower than in COVID-19-related preprints shared via medRxiv and bioRxiv [26]. While some research articles may not be able to share their data due to constraints such as privacy or legal obligations, at least most of them could have shared their metadata, that is, "descriptive information about the context, quality and condition, or characteristics of the data" including "e.g., the data captured automatically by machines that generate data such as

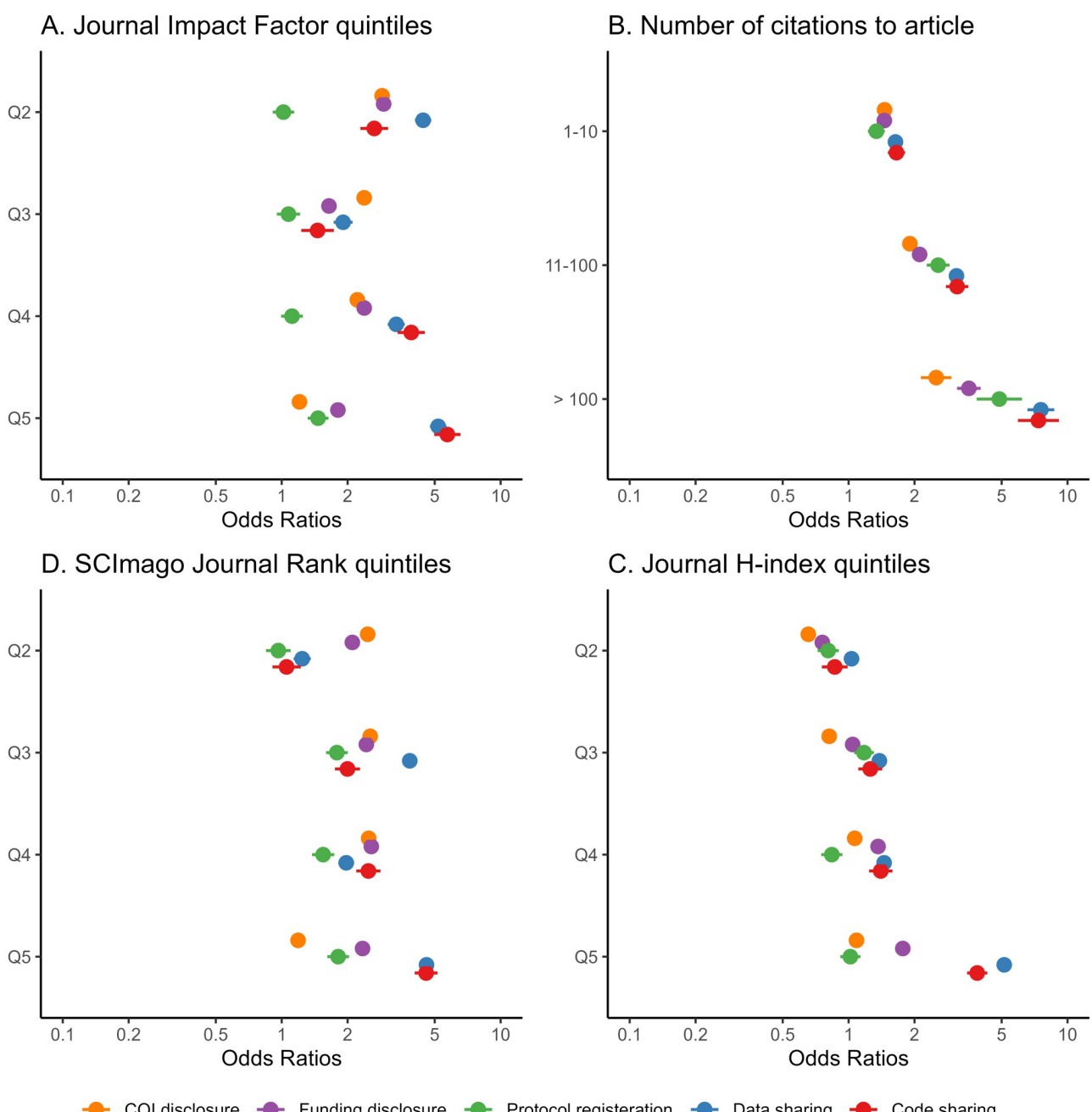

**Fig 4. Transparency practices in research articles by Journal Impact Factor quintiles (A), number of citations to article (B), SCImago Journal Rank quintiles (C) and journal H-index quintiles (D).** Odds ratios were adjusted for month-year of article publication and whether the article was a randomized controlled trial.

DICOM information for image files" [27]. On the other hand, a recent study of 200 articles showed that adherence to COI and funding disclosures improved from preprints to peer-reviewed publications and showed higher adherence to funding disclosures than our findings indicated [28]. In addition, adherence to transparency practices was higher than in our previous study on COVID-19-related research in dental journals [21].

It has been highlighted that transparency practices are associated with higher impact, that is, citations [29]. Even though meta-research findings have not always been that clear [9]. For instance, in the study of Serghiou et al. [9] with the same methodology as here, COI disclosure was associated with a slightly lower number of citations (7 IQR: 3–18 vs. 6 IQR: 2–14). According to our findings, the relationship between the number of received citations and transparency indicators was stronger and more homogenous than the relationship between journal impact/rank/H-index and transparency indicators, and stronger than that found by Serghiou et al. [9]. The relationships between transparency practices and the Transparency and Openness (TOP) Factor of journals could be explored in the future (https://topfactor.org/).

Even though we used representative data and validated methods to investigate transparency in COVID-19-related research, our study has some weaknesses. First, with the applied methods, we could not accurately distinguish all the studies required to register their protocol, e.g., distinguishing meta-analyses from narrative reviews or studies that did not produce any data or code to share. So this means that even though every researcher would have adhered to all five practices, we would not likely have achieved 100% adherence rates (e.g., because not every study uses any statistical procedures). Also, the study sample was restricted to open access articles in the EPMC database, which may not correspond to all COVID-19-related studies published in medical journals. At least, in general, the differences in transparency practices between open access and non-open access articles from EPMC are similar [30]. However, as most COVID-19-related articles were available via EPMC (73%), this does not likely diminish the strength of our interpretations considerably. In addition, our analyses were restricted to the published articles. Thus, we did not evaluate the material provided or shared during peer review, which may have been more comprehensive than what ended up in the published article.

Transparency is crucial to ensure the credibility of science and enable its assessment [23, 24]. Transparent scientific practices, like the ones we investigated here, have helped to fight the pandemic globally [6, 7]. While most COVID-19-related articles were open access and adhered to disclosing funding and COI, our findings showed suboptimal adherence to data, code sharing, and protocol registration. A stronger and more concrete commitment to open science practices, particularly to protocol registration, data, and code sharing, is needed from all stakeholders. societies and their people would be the beneficiary [23].

## Supporting information

**S1 Text. Deviations from the protocol.**
(DOCX)

**S1 Table. Citations to article and journal impact factor by transparency practices for RCTs.**
(DOCX)

**S2 Table. Citations to article and journal impact factor by transparency practices for reviews.**
(DOCX)

**S3 Table. SCImago Journal Rank and H-index by transparency practices for RCTs.**
(DOCX)

**S4 Table. SCImago Journal Rank and H-index by transparency practices for reviews.**
(DOCX)

**S1 Appendix. Random sample and its validation.**
(CSV)

## Author Contributions

**Conceptualization:** Ahmad Sofi-Mahmudi, Eero Raittio, Sergio E. Uribe.

**Data curation:** Ahmad Sofi-Mahmudi, Eero Raittio.

**Formal analysis:** Ahmad Sofi-Mahmudi, Eero Raittio.

**Investigation:** Ahmad Sofi-Mahmudi, Eero Raittio.

**Methodology:** Ahmad Sofi-Mahmudi, Eero Raittio.

**Project administration:** Ahmad Sofi-Mahmudi, Eero Raittio.

**Resources:** Ahmad Sofi-Mahmudi, Eero Raittio.

**Software:** Ahmad Sofi-Mahmudi, Eero Raittio.

**Supervision:** Sergio E. Uribe.

**Validation:** Ahmad Sofi-Mahmudi, Eero Raittio.

**Visualization:** Ahmad Sofi-Mahmudi, Eero Raittio.

**Writing – original draft:** Ahmad Sofi-Mahmudi, Eero Raittio.

**Writing – review & editing:** Ahmad Sofi-Mahmudi, Eero Raittio, Sergio E. Uribe.

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
