## [Decision Letter · Decision Letter 0]

14 Feb 2023

PONE-D-22-33929Transparency of COVID-19-related research: A meta-research studyPLOS ONE

Dear Ahmad Sofi-Mahmudi, DDS

Thank you for submitting your manuscript to PLOS ONE. After careful consideration, We feel that it has merit but does not fully meet PLOS ONE’s publication criteria as it currently stands. Therefore, we invite you to submit a revised version of the manuscript that addresses the points raised during the review process. The paper deals with the issue of transparency about the research generated in terms of derived scientific publications, especially after having experienced the Covid-19 pandemic. This has highlighted the need to quickly communicate the results generated, with transparency to ensure their reproducibility and facilitate their reuse.

First, I would like to thank the 3 reviewers for their important comments. I have the following additional comments:

- The introduction should better explain open science and open data scenario with data sharing practices (data deposited in repositories or data from supplementary material, etc.), as well as code sharing. It is about transparency but no comments on FAIR principles and research data are done even in discussion.

- Related to the methods: Document search strategy needs better description. As I understood you have searched for Covid-19 publications in PubMed database. At this point which are the terms used for the search equation?

-Also you have used LitCovid for searching articles and reviews and L.OVE platform for searching clinical trials, is that correct? With the merged PMIDs that you have obtained from the PubMed, LitCovid and L.OVE databases, a second search has been performed in Europe PubMed Central to obtain the full texts of the papers (can be difficult to understand even reading S1 text-deviations from the protocol).

Subsequently an analysis of the total papers obtained from EPMC has been done for 5 transparency practices (and results are showed by type of document: research article, review or RCT). All this search strategy should be summarized at the beginning of the results to improve readers' understanding. Even a flow chart with the data obtained (database and number or results) would be interesting and would facilitate understanding.

- The authors need to discuss their choice of database--and the fact that they only searched one database--as a limitation of their research. Why EPMC? Why not PMC for example?

In the methodology, the tool that has been used should be better explained, because the manual validation that we can see in the supplementary excel is only for 100 documents.

-Table 5 and supplementary tables needs to clarify. It is needed an explanation regarding the significance of Median (IQR) with/without.

- In addition to the questions previously asked, as suggested by the 3 reviewers, please respond to each point that they have questioned.

As far as I can see, I cannot promise acceptance, but I would reconsider an improved version of the current manuscript.

Reviewer 1

The subject matter of the work presented is of great importance to the scientific community. Transparency is one of the fundamental pillars on which most emphasis is currently being placed by international organizations and funding agencies.

In my opinion, I believe that the article is interesting and useful for the scientific community.

However, it has some shortcomings that I believe should be addressed.

1. The paper mentions that the five practices that underpin transparency are data sharing, code sharing, COI disclosures, and protocol registration. Considering that not everyone is familiar with the concepts, I think a brief description of each in the enumeration of the concepts in the "Data extraction and synthesis" section of the methodology would be very useful. Based on this description, I believe that it should be further specified what is to be considered in this study for each element. This is particularly important in the case of data sharing since within this group it is not differentiated whether it is raw data or already processed material, as can be seen in the S7 Appendix for example in the open data statement of record 103787, which is not raw data.

2. On the other hand, I think that, in addition to documentary typology, it would be very interesting to add some information on the distribution at least by country or by discipline, as I think it would give a richer context to the study. The subject matter of COVID-19 is very broad and the disciplines that deal with it in one way or another are very diverse. It would be very useful to know which disciplines are carrying out transparency practices and how, for example, which ones are sharing more code, or which ones are sharing more data. This type of information would enrich the discussion, and more robust conclusions could be drawn, since as they stand they are quite generic and highlight aspects that are not novel. In other words, we already know that more transparency is crucial to ensure credibility and that society should benefit from it, but at this point, it would be interesting to sharpen the focus and find out, within the medical disciplines, who does what and how, since genetics and its tradition of sharing data (regardless of covid) have nothing to do with more clinical disciplines.

Reviewer 2

This is a very descriptive study in which the results of other studies are commented on without a clear relationship with the study addressed in this work.

Method

1.The authors rely on the results of the Europe PubMed Central database. Explain the reasons why they have chosen this source over others, as well as the possible biases that this may have produced.

2.Please, explain how do you measure the data sharing variable?

3.Describe what the tool used considers “valid” or considers dataset. Explain what the tool consists of. It is not enough just to cite it in the bibliography. For example, does it include supplementary material or just datasets?

Results.

4.In this chapter it is striking that nothing is said about the journals in Table 3 and the publishers in Table 4.

5.Table 5 only includes citations and impact factor. Why is the h-index and the number of articles published in open access not included?

Discussion

6.The discussion is poor in arguments and repeats again some of the results of the study. It seems to be unfinished or halfway through.

7.Some aspects to be considered and argued have to do with transparency, since transparency alone is not synonymous with quality.

Authors should at least relate transparency practices to other indicators of article quality, such as the number of citations received, impact factor, journal citation Indicator and quartile, since, on the other hand, it has been mentioned in the method.

8.The fact that review articles receive less funding is obvious, since this type of work is not usually funded, as they are not research work sensu stricto.

9.What are the implications of the fact that "COI disclosures seemed to be more common in articles published in lower-impact journals?

10.Why "adherence to transparent practices was at a similar level in COVID-19-related literature than in other biomedical literature analyzed with the same methods"?

References

What do you mean by "Scientometrics. 2021 kesäkuu"?

Reviewer 3

This article deals with the concept of transparency in works (research, RCT, reviews) published in PubMed indexed journals about COVID-19.

My major concerns focus the lack of explanations.

1. The background section is too short, and it is excessively centered in transparency, omitting that concepts like data sharing go beyond transparency. Authors do not explain what is data sharing or code sharing and why they are so important for transparency, among other considerations.

2.The Methods section lacks of an explanation about why Europe PMC repository was selected, despite of the potential problems about this decision were included in the Discussion section.

3.Table 5 is not sufficiently understandable.

We look forward to receiving your revised manuscript.

Kind regards,

Rut Lucas-Dominguez, PhD

Academic Editor

PLOS ONE

Journal Requirements:

Important: If there are ethical or legal restrictions to sharing your data publicly, please explain these restrictions in detail. Please see our guidelines for more information on what we consider unacceptable restrictions to publicly sharing data: http://journals.plos.org/plosone/s/data-availability#loc-unacceptable-data-access-restrictions  Note that it is not acceptable for the authors to be the sole named individuals responsible for ensuring data access.

Reviewers' comments:

Reviewer's Responses to Questions

**Comments to the Author**

1. Is the manuscript technically sound, and do the data support the conclusions?

Reviewer #1: Yes

Reviewer #2: Yes

Reviewer #3: Yes

2. Has the statistical analysis been performed appropriately and rigorously? 

Reviewer #1: Yes

Reviewer #2: Yes

Reviewer #3: Yes

3. Have the authors made all data underlying the findings in their manuscript fully available?

Reviewer #1: Yes

Reviewer #2: Yes

Reviewer #3: Yes

4. Is the manuscript presented in an intelligible fashion and written in standard English?

Reviewer #1: Yes

Reviewer #2: Yes

Reviewer #3: Yes

5. Review Comments to the Author

Reviewer #1: The subject matter of the work presented is of great importance to the scientific community. Transparency is one of the fundamental pillars on which most emphasis is currently being placed by international organizations and funding agencies.

In my opinion, I believe that the article is interesting and useful for the scientific community.

However, it has some shortcomings that I believe should be addressed.

The paper mentions that the five practices that underpin transparency are data sharing, code sharing, COI disclosures, and protocol registration. Considering that not everyone is familiar with the concepts, I think a brief description of each in the enumeration of the concepts in the "Data extraction and synthesis" section of the methodology would be very useful. Based on this description, I believe that it should be further specified what is to be considered in this study for each element. This is particularly important in the case of data sharing since within this group it is not differentiated whether it is raw data or already processed material, as can be seen in the S7 Appendix for example in the open data statement of record 103787, which is not raw data.

On the other hand, I think that, in addition to documentary typology, it would be very interesting to add some information on the distribution at least by country or by discipline, as I think it would give a richer context to the study. The subject matter of COVID-19 is very broad and the disciplines that deal with it in one way or another are very diverse. It would be very useful to know which disciplines are carrying out transparency practices and how, for example, which ones are sharing more code, or which ones are sharing more data. This type of information would enrich the discussion, and more robust conclusions could be drawn, since as they stand they are quite generic and highlight aspects that are not novel. In other words, we already know that more transparency is crucial to ensure credibility and that society should benefit from it, but at this point, it would be interesting to sharpen the focus and find out, within the medical disciplines, who does what and how, since genetics and its tradition of sharing data (regardless of covid) have nothing to do with more clinical disciplines.

Reviewer #2: PONE-D-22-33929

Transparency of COVID-19-related research: A meta-research study

This is a very descriptive study in which the results of other studies are commented on without a clear relationship with the study addressed in this work.

Method

The authors rely on the results of the Europe PubMed Central database. Explain the reasons why they have chosen this source over others, as well as the possible biases that this may have produced.

Please, explain how do you measure the data sharing variable?

Describe what the tool used considers “valid” or considers dataset. Explain what the tool consists of. It is not enough just to cite it in the bibliography. For example, does it include supplementary material or just datasets?

Results.

In this chapter it is striking that nothing is said about the journals in Table 3 and the publishers in Table 4.

Table 5 only includes citations and impact factor. Why is the h-index and the number of articles published in open access not included?

Discussion

The discussion is poor in arguments and repeats again some of the results of the study. It seems to be unfinished or halfway through.

Some aspects to be considered and argued have to do with transparency, since transparency alone is not synonymous with quality.

Authors should at least relate transparency practices to other indicators of article quality, such as the number of citations received, impact factor, journal citation Indicator and quartile, since, on the other hand, it has been mentioned in the method.

The fact that review articles receive less funding is obvious, since this type of work is not usually funded, as they are not research work sensu stricto.

What are the implications of the fact that "COI disclosures seemed to be more common in articles published in lower-impact journals?

Why "adherence to transparent practices was at a similar level in COVID-19-related literature than in other biomedical literature analyzed with the same methods"?

References

What do you mean by "Scientometrics. 2021 kesäkuu"?

Reviewer #3: This article deals with the concept of transparency in works (research, RCT, reviews) published in PubMed indexed journals about COVID-19.

My major concerns focus the lack of explanations. The background section is too short, and it is excessively centered in transparency, omitting that concepts like data sharing go beyond transparency. Authors do not explain what is data sharing or code sharing and why they are so important for transparency, among other considerations.

The Methods section lacks of an explanation about why Europe PMC repository was selected, despite of the potential problems about this decision were included in the Discussion section.

Table 5 is not sufficiently understandable.

6. PLOS authors have the option to publish the peer review history of their article (what does this mean?). If published, this will include your full peer review and any attached files.

Reviewer #1: No

Reviewer #2: No

Reviewer #3: No

---

## [Author Response · Author response to Decision Letter 0]

18 Apr 2023

Associate Editor comments:

- The introduction should better explain open science and open data scenario with data sharing practices (data deposited in repositories or data from supplementary material, etc.), as well as code sharing. It is about transparency but no comments on FAIR principles and research data are done even in discussion.

Authors: Thanks for your comment. We revised the whole introduction part, and it now includes all the topics that you mentioned.

- Related to the methods: Document search strategy needs better description. As I understood you have searched for Covid-19 publications in PubMed database. At this point which are the terms used for the search equation?

-Also you have used LitCovid for searching articles and reviews and L.OVE platform for searching clinical trials, is that correct? With the merged PMIDs that you have obtained from the PubMed, LitCovid and L.OVE databases, a second search has been performed in Europe PubMed Central to obtain the full texts of the papers (can be difficult to understand even reading S1 text-deviations from the protocol).

Subsequently an analysis of the total papers obtained from EPMC has been done for 5 transparency practices (and results are showed by type of document: research article, review or RCT). All this search strategy should be summarized at the beginning of the results to improve readers' understanding. Even a flow chart with the data obtained (database and number or results) would be interesting and would facilitate understanding.

- The authors need to discuss their choice of database--and the fact that they only searched one database--as a limitation of their research. Why EPMC? Why not PMC for example?

Authors: Thanks for your valuable comment. We revised the Methods and the Results sections accordingly and added a Venn diagram and a flow diagram. In summary, we first searched for all the papers indexed in PubMed via Europe PMC (EPMC) by just specifying date (FIRST_PDATE:[2020-01-01 TO 2022-06-09]), the database of interest (SRC:"MED"), open-access status (OPEN_ACCESS:y), and English language (LANG:"eng" OR LANG:"en" OR LANG:"us") without any keywords. The EPMC includes all the articles indexed in several databases, including PubMed and PubMed Central (PMC). We chose EPMC because it has capabilities to be used for Open Science research (e.g., europepmc package in R). More information is available here: https://europepmc.org/About.

After catching over 1.5 million articles, we used the LitCovid database to check which of these articles were COVID-19-related. PubMed (and hence, EPMC) have filters for recognizing publication types (such as research articles, letters, reviews, RCTs, observational studies, etc.). However, based on our investigation, only research article and review labels are accurate, and many RCTs and observational studies are labelled solely as research articles rather than RCTs or observational studies. Therefore, we used the L.OVE database to detect COVID-19-related RCTs. Unfortunately, we did not find any database for detecting observational studies. 

In the methodology, the tool that has been used should be better explained, because the manual validation that we can see in the supplementary excel is only for 100 documents.

Authors: Thanks for your comment. We revised the Methods section with complementary information in this regard. This tool has been previously validated for all biomedical research papers (https://doi.org/10.1371/journal.pbio.3001107). We performed the validation exercise to investigate whether this tool is still valid for COVID-19-related papers.

-Table 5 and supplementary tables needs to clarify. It is needed an explanation regarding the significance of Median (IQR) with/without.

Authors: Thanks for your comments. We added a thorough explanation and also moved Table S4 to the main text as Table 6 since we thought it also could add more insights to our results.

- In addition to the questions previously asked, as suggested by the 3 reviewers, please respond to each point that they have questioned.

As far as I can see, I cannot promise acceptance, but I would reconsider an improved version of the current manuscript.

Authors: Thanks for all your valuable comments. We think your comments and those of the reviewers improved our manuscript substantially.

 

Reviewer 1

The subject matter of the work presented is of great importance to the scientific community. Transparency is one of the fundamental pillars on which most emphasis is currently being placed by international organizations and funding agencies.

In my opinion, I believe that the article is interesting and useful for the scientific community.

However, it has some shortcomings that I believe should be addressed.

1. The paper mentions that the five practices that underpin transparency are data sharing, code sharing, COI disclosures, and protocol registration. Considering that not everyone is familiar with the concepts, I think a brief description of each in the enumeration of the concepts in the "Data extraction and synthesis" section of the methodology would be very useful. Based on this description, I believe that it should be further specified what is to be considered in this study for each element. This is particularly important in the case of data sharing since within this group it is not differentiated whether it is raw data or already processed material, as can be seen in the S7 Appendix for example in the open data statement of record 103787, which is not raw data.

Authors: Thanks for your valuable comment. We amended the methods section accordingly, with examples for more elaboration. 

2. On the other hand, I think that, in addition to documentary typology, it would be very interesting to add some information on the distribution at least by country or by discipline, as I think it would give a richer context to the study. The subject matter of COVID-19 is very broad and the disciplines that deal with it in one way or another are very diverse. It would be very useful to know which disciplines are carrying out transparency practices and how, for example, which ones are sharing more code, or which ones are sharing more data. This type of information would enrich the discussion, and more robust conclusions could be drawn, since as they stand they are quite generic and highlight aspects that are not novel. In other words, we already know that more transparency is crucial to ensure credibility and that society should benefit from it, but at this point, it would be interesting to sharpen the focus and find out, within the medical disciplines, who does what and how, since genetics and its tradition of sharing data (regardless of covid) have nothing to do with more clinical disciplines.

Authors: Thanks for your thoughtful insight. We agree that this is very important. However, do not have the data for countries or discipline. In fact, we aim to do so in our next project for all 59 categories of clinical medicine. For such a research, we should have a list of journals in each filed of health and medicine and the countries of the authors. Unfortunately, Europe PMC and PubMed do not share such data when returning the search results. Therefore, some alternative methods such as using categorization of the other databases (e.g., JCR or SJR) should be used at the first stage (searching for the articles). To sum up, we have constraints to do so with the current research.

 

Reviewer 2

This is a very descriptive study in which the results of other studies are commented on without a clear relationship with the study addressed in this work.

Method

1.The authors rely on the results of the Europe PubMed Central database. Explain the reasons why they have chosen this source over others, as well as the possible biases that this may have produced.

Authors: Thanks for your comment. MEDLINE (and hence, PubMed) is the most important database of articles published in health and medical journals. The Europe PMC (EPMC) has exactly the same database as MEDLINE and PubMed. As they have mentioned in their About page (https://europepmc.org/About), it includes PubMed and PubMed Central (PMC) and more than 30 other sources. In this research, we just used the PubMed part of it. The reason for using EPMC was that it enabled us to do all the works automatically which PubMed website is not capable of doing so. The results of the search queries in EPMC and PubMed are exactly the same and, therefore, we can tell that we used PubMed (or MEDLINE) through EPMC (just like using MEDLINE through Ovid).

We added a sentence for the clarification under the “Data sources and study selection” section of the Methods which reads:

“This database includes all the records available through PubMed and PubMed Central and also enabled us to retrieve the record automatically which is not available through PubMed website.”

2.Please, explain how do you measure the data sharing variable?

Authors: Thanks for the comment. We amended the “Data extraction and synthesis” section accordingly. Now it reads:

“We used a validated and automated tool developed by Serghiou et al. (6) suitable to identify these five transparent practices from articles in XML format from the EPMC database. Briefly, this tool uses some keywords to identify adherence to each indicator using regular expressions. For example, it searches for phrases commonly associated with a COI disclosure (e.g., “conflicts of interest,” “competing interests,” “Nothing to disclose.,” etc.) in the body or titles of the sections of the text file of an article. For data and code sharing detection, it both detects shared as a supplement or shared in a general (e.g., figshare, OSF, GitHub, etc.) or field-specific repository (e.g., dbSNP, ProteomeXchange, GenomeRNAi, etc.) as adherence to transparency to data/code sharing. However, those articles that indicated "data available under request" would be classified as no data available, since it is unlikely to obtain these data (11). Overall, all the mentions of the indicators were classified as affirmative when the article explicitly contained it.”

3.Describe what the tool used considers “valid” or considers dataset. Explain what the tool consists of. It is not enough just to cite it in the bibliography. For example, does it include supplementary material or just datasets?

Authors: Thanks for the comment. As we mentioned above, we elaborated more on the algorithm of the tool.

Results.

4.In this chapter it is striking that nothing is said about the journals in Table 3 and the publishers in Table 4.

Authors: Thanks for your keen comment. We aimed to just include the most important aspects of these tables (such as “55% of research articles published in PLoS One had shared data”). However, we extended this paragraph to show more interesting findings of these Tables. Now it reads:

“There were journal and publisher-related differences in all five practices (Tables 3 and 4). All the papers from MDPI and PLoS had COI and funding disclosure. Whereas COI and funding disclosure was available for almost all the RCTs published in the New England Journal of Medicine and The Lancet, they rarely shared their data (0% and 2.9%, respectively). The highest percentage of data sharing was found in PLoS One research articles (54.8%) and RCTs (70.0%). RCTs published in The Lancet and The Lancet Respiratory Medicine had the highest adherence to protocol registration, with 94.1% and 100%, respectively. Code sharing was a rare practice, with Scientific Reports (14.2%) and PLoS (11.1%) be the top journal and publisher for adherence in this practice. None of the journals with the highest number of RCTs in the sample shared their codes.”

5.Table 5 only includes citations and impact factor. Why is the h-index and the number of articles published in open access not included?

Authors: Thanks for your comment. We transferred Table 6 from Appendix S4 to the main text to show the results for H-index and SCImago Journal Rank (SJR). We also added the following paragraph:

“Table 6 illustrates the median SCImago Journal Rank (SJR) and H-index for articles with and without adherence to each transparency indicator. Research articles that adhered to each of the five indicators had higher median SJR and H-index. The results for RCTs and reviews are available in Tables S4 and S5.”

Discussion

6.The discussion is poor in arguments and repeats again some of the results of the study. It seems to be unfinished or halfway through.

Authors: Thanks for the comment. We extended the discussion and amended it considerably.

7.Some aspects to be considered and argued have to do with transparency, since transparency alone is not synonymous with quality.

Authors should at least relate transparency practices to other indicators of article quality, such as the number of citations received, impact factor, journal citation Indicator and quartile, since, on the other hand, it has been mentioned in the method.

Authors: Thanks for your valuable comment. We had mentioned article quality indices in the discussion very briefly: “differences in transparent practices according to received citations and JIF were small”. We elaborated on this in a paragraph, as below:

“Whereas transparency is deemed to be important for health research (PMID: 26821973), when we compared the level of transparency for each domain based on some indices related to the quality of the papers and journals (Tables 5 and 6), we found inconsistent results. For example, there were little differences between articles adhered to transparency indicators in terms of citation. The same was seen for SJR index. However, journals with higher JIF and H-index had better adherence to transparency indicators. This may imply that publishers pay attention to the trends and have updated their policy regarding transparency, especially in COI and funding disclosure. However, researchers cite papers without considering their level of transparency.”

8.The fact that review articles receive less funding is obvious, since this type of work is not usually funded, as they are not research work sensu stricto.

Authors: Thanks for your keen insight. In fact, the tool that we used just determines mentioning of funding disclosure and does not differentiate between articles that received funding with those which did not. In other words, if there was a sentence such as “no funding” in the paper, that paper will be considered as transparent in terms of funding disclosure. However, for clearance, we added examples in the “Data extraction and synthesis” section of the Methods.

Furthermore, the adherence of reviews to transparency indicators is comparable to research articles as depicted in Table 1).

9.What are the implications of the fact that "COI disclosures seemed to be more common in articles published in lower-impact journals?

Authors: Thanks for the comment. That sentence was not accurate based on Table 5. We amended it as below:

“In addition, journals with higher JIF seemed to be more transparent, except for the COI disclosures”.

Other explanations are available in the paragraph added for your 7th comment.

10.Why "adherence to transparent practices was at a similar level in COVID-19-related literature than in other biomedical literature analyzed with the same methods"?

Authors: Thanks for the comment. We added two sentence to the end of that paragraph for possible reasons:

“Furthermore, any initiative and movement needs time to be effective. Another reason could be the fact that these changes should be implemented by publishers and journals.”

References

What do you mean by "Scientometrics. 2021 kesäkuu"?

Authors: Thanks for your keen comment. This is Finnish for “June”. We substituted it with “June”.

 

Reviewer 3

This article deals with the concept of transparency in works (research, RCT, reviews) published in PubMed indexed journals about COVID-19.

My major concerns focus the lack of explanations.

1. The background section is too short, and it is excessively centered in transparency, omitting that concepts like data sharing go beyond transparency. Authors do not explain what is data sharing or code sharing and why they are so important for transparency, among other considerations.

Authors: Thanks for your comment. This concern was also raised by other reviewers. We amended all the comments regarding the Background.

2.The Methods section lacks of an explanation about why Europe PMC repository was selected, despite of the potential problems about this decision were included in the Discussion section.

Authors: Thanks for the comment. MEDLINE (and hence, PubMed) is the most important database of articles published in health and medical journals. The Europe PMC (EPMC) has exactly the same database as MEDLINE and PubMed. As they have mentioned in their About page (https://europepmc.org/About), it includes PubMed and PubMed Central (PMC) and more than 30 other sources. In this research, we just used the PubMed part of it. The reason for using EPMC was that it enabled us to do all the works automatically which PubMed website is not capable of doing so. The results of the search queries in EPMC and PubMed are exactly the same and, therefore, we can tell that we used PubMed (or MEDLINE) through EPMC (just like using MEDLINE through Ovid).

We added a sentence for the clarification under the “Data sources and study selection” section of the Methods which reads:

“This database includes all the records available through PubMed and PubMed Central and also enabled us to retrieve the record automatically which is not available through PubMed website.”

We also mentioned the limitations of this decision in the discussion.

3.Table 5 is not sufficiently understandable.

Authors: Thanks for your comment. We added more details for that part and also added Table 6 which includes SCImago Journal Rank (SJR) and H-index. Now it reads:

“Apart from COI disclosure, research articles that adhered to transparency practices were published in journals with higher median JIF. Research articles with data or code sharing received one more median citation than those without data or code sharing. In contrast, articles without COI or funding disclosure received one more median citation compared to those which adhered to these indicators (Table 5). Tables S2 and S3 show the results for RCTs and reviews.”

“Table 6 illustrates the median SCImago Journal Rank (SJR) and H-index for articles with and without adherence to each transparency indicator. Research articles that adhered to each of the five indicators had higher median SJR and H-index. The results for RCTs and reviews are available in Tables S4 and S5.”

---

## [Editor Report · Decision Letter 1]

18 May 2023

PONE-D-22-33929R1Transparency of COVID-19-related research: A meta-research studyPLOS ONE

Dear Dr. Sofi-Mahmudi,

Thank you for submitting your manuscript to PLOS ONE. After careful consideration, we feel that it has merit but does not fully meet PLOS ONE’s publication criteria as it currently stands. Therefore, we invite you to submit a revised version of the manuscript that addresses the points raised during the review process. Please submit your revised manuscript by Jul 02 2023 11:59PM. If you will need more time than this to complete your revisions, please reply to this message or contact the journal office at plosone@plos.org. Please include the following items when submitting your revised manuscript:A rebuttal letter that responds to each point raised by the academic editor. You should upload this letter as a separate file labeled 'Response to Reviewers'.A marked-up copy of your manuscript that highlights changes made to the original version. You should upload this as a separate file labeled 'Revised Manuscript with Track Changes'.An unmarked version of your revised paper without tracked changes. You should upload this as a separate file labeled 'Manuscript'.If applicable, we recommend that you deposit your laboratory protocols in protocols.io to enhance the reproducibility of your results. Protocols.io assigns your protocol its own identifier (DOI) so that it can be cited independently in the future. For instructions see: https://journals.plos.org/plosone/s/submission-guidelines#loc-laboratory-protocols. Additionally, PLOS ONE offers an option for publishing peer-reviewed Lab Protocol articles, which describe protocols hosted on protocols.io. Read more information on sharing protocols at https://plos.org/protocols?utm_medium=editorial-email&utm_source=authorletters&utm_campaign=protocols.

We look forward to receiving your revised manuscript.

Kind regards,

Rut Lucas-Dominguez, PhD

Academic Editor

PLOS ONE

Journal Requirements:

**Additional Editor Comments:**

The paper has been improved following the recommendations of the academic editor and reviewers. It is important for the acceptation of the manuscript that the minor changes suggested will be taken into account.

1. Background. There are spaces in the text before citations 3 and 4 that need to be removed.

2. Background. The sentence: To facilitate rapid and collaborative scientific research, over 100 organizations…please could authors insert the reference to support this sentence.

3. Methods. At the end of Data sources and study selection it is mentioned: As we want to focus solely on RCTs that have been done, we applied two filters: “RCT” and “Reporting data,” on the L·OVE website and then downloaded the dataset with these characteristics. Again, we used PMIDs to merge datasets. Please, check if is correct that sentence? The PMIDs are used to recover the articles?, not datasets. In fact, authors are explaining in the next paragraph of Data extraction and synthesis: we assessed articles’ adherence to five transparent practices.

4. Methods. Regarding journal and article information in METHODS, please explain what IQM is and how is calculated. It appears in the tables of results many times.

5. Methods. In the table 5 Where did the citations come from? Are calculated through the Web of Science, since is the same table of JIF. Explain please. To clarify, authors may include citations and Impact index of the journals were obtained from JCR. Which edition of JCR are you using? Mainly because for the other impact indicators in the following table you are using SJR and h-index from another database (SJR).

6. Methods. Data analysis. Regarding the tables of results, appears in the legend of the tables SD: standard deviation. But never appears this acronym in the tables, so this may be confusing.

7. Results. Transparency practices. It appears figure 1, however authos have to correct writing figure 3, as well as Figure 3-right.

8. Results. At the end of table 4: Tables S2 and S3 show the results for RCTs and reviews.

I can not see this supplementary tables. I only can see Table S4, Table S5 and S6 appendix in the submission files.

9. Results. Table 5. SD appears in the explanatory legend below the table: standard deviation. But this acronym does not appear in the table. For example, IQR appears in the table, and it is explained as inter quartile range. At this point, please explain the meaning/calculation of Median inter-quartile range (IQR) in methodology.

10. Results. Of 100 random articles for validation, 20 discrepancies (out of 500, 4%), could authors explain the meaning of: out of 500, 4%.

11. Results. (S7 Appendix). I cannot see S7 appendix.

12. Discussion. Due to the manuscript has many tables, it would be great if the authors could synthetize in brief short lines at the beginning of the discussion the results obtained.Please insert comments here and delete this placeholder text when finished.

---

## [Author Response · Author response to Decision Letter 1]

18 May 2023

Journal Requirements:

Authors: We reviewed all the references.

Additional Editor Comments:

The paper has been improved following the recommendations of the academic editor and reviewers. It is important for the acceptation of the manuscript that the minor changes suggested will be taken into account.

1. Background. There are spaces in the text before citations 3 and 4 that need to be removed.

Authors: Thanks for your keen look. We removed these spaces and also some others. We used Google Docs for the previous revision, and that caused some deformities in the text. For this revision, we used Microsoft Word, and we hope it works fine this time.

2. Background. The sentence: To facilitate rapid and collaborative scientific research, over 100 organizations…please could authors insert the reference to support this sentence.

Authors: We added the pertaining reference which is a press release from Wellcome Trust: https://wellcome.org/press-release/sharing-research-data-and-findings-relevant-novel-coronavirus-ncov-outbreak

3. Methods. At the end of Data sources and study selection it is mentioned: As we want to focus solely on RCTs that have been done, we applied two filters: “RCT” and “Reporting data,” on the L·OVE website and then downloaded the dataset with these characteristics. Again, we used PMIDs to merge datasets. Please, check if is correct that sentence? The PMIDs are used to recover the articles?, not datasets. In fact, authors are explaining in the next paragraph of Data extraction and synthesis: we assessed articles’ adherence to five transparent practices.

Authors: Thanks for the comment. We clarified the sentence. Now it reads:

“We used PMIDs of COVID-19-related RCTs provided in this downloaded dataset from the L·OVE website to detect RCTs in our main dataset of all open-access COVID-19-related papers.”.

More clarification: The LOVE website allows to download a dataset of articles with desired characteristics (COVID-19-related RCTs in our case). Then, we used the intersection between this dataset and our dataset of all COVID-19-related papers to detect open-access COVID-19-related RCTs.

4. Methods. Regarding journal and article information in METHODS, please explain what IQM is and how is calculated. It appears in the tables of results many times.

Authors: Thanks for the comment. We clarified it in the last paragraph of methods:

“We used the interquartile range (the third quartile (Q3) – the first quartile (Q1)) when the data were skewed.”.

5. Methods. In the table 5 Where did the citations come from? Are calculated through the Web of Science, since is the same table of JIF. Explain please. To clarify, authors may include citations and Impact index of the journals were obtained from JCR. Which edition of JCR are you using? Mainly because for the other impact indicators in the following table you are using SJR and h-index from another database (SJR).

Authors: Thanks for the comment. As we have indicated in the last paragraph of the “Data extraction and synthesis”, it comes from PubMed (through EPMC database):

“Basic journal- and article-related information (publisher, publication year, citations to article and journal name) were retrieved from the EPMC database.”

Also, in the same paragraph, the source of JIF is indicated:

“The impact index of the journals (JIFs) was obtained from the Journal Citation Reports 2021 (jcr.clarivate.com) and the SCImago Journal Rank 2020 (SJR) and H-index indicators from the SCImago website (scimagojr.com).”

We added the editions to the text (2021 and 2020, respectively).

6. Methods. Data analysis. Regarding the tables of results, appears in the legend of the tables SD: standard deviation. But never appears this acronym in the tables, so this may be confusing.

Authors: Thanks for your keen comment. We removed the SD.

7. Results. Transparency practices. It appears figure 1, however authos have to correct writing figure 3, as well as Figure 3-right.

Authors: Thanks for your comment. We carefully revised all the figures and legends and references to them in the text.

8. Results. At the end of table 4: Tables S2 and S3 show the results for RCTs and reviews.

I can not see this supplementary tables. I only can see Table S4, Table S5 and S6 appendix in the submission files.

Authors: We think there has been some mistake when submitting the previous revision. Now you should receive Tables S2 and S3 files.

9. Results. Table 5. SD appears in the explanatory legend below the table: standard deviation. But this acronym does not appear in the table. For example, IQR appears in the table, and it is explained as inter quartile range. At this point, please explain the meaning/calculation of Median inter-quartile range (IQR) in methodology.

Authors: Thanks for the comment. As we mentioned in the response for comments 4 and 6, we added the explanation for IQR and removed SD from all the tables.

10. Results. Of 100 random articles for validation, 20 discrepancies (out of 500, 4%), could authors explain the meaning of: out of 500, 4%.

Authors: Thanks for the comment. Because every articles is being assessed for 5 indicators, we will have 5x100=500 indicators overall. Of these, 20 (4%) had discrepancies. We added “five indicators per article” The paragraph now reads:

“Of 100 random articles for validation, 20 discrepancies (out of 500 (five indicators per article), 4%) between the automatic tool and manual checking were found: 9% for open data, 2% for open code, 1% for COI disclosure, 6% for funding disclosure, and 2% for registration (S7 Appendix).”

11. Results. (S7 Appendix). I cannot see S7 appendix.

Authors: We think there has been a mistake in submitting the appendices. We carefully checked that all the appendices are included in this submission.

12. Discussion. Due to the manuscript has many tables, it would be great if the authors could synthetize in brief short lines at the beginning of the discussion the results obtained.

Authors: Thanks for the suggestion. We added an extra summary at the beginning of the discussion. Now it reads:

“Our study showed that adherence to transparent practices increased in COVID-19-related medical literature from 2020 to 2022. Adherence to reporting COI disclosure was high throughout the study period. In addition, most articles had funding disclosure. Data sharing, code sharing and protocol registration were rare but improved little over the study period. Higher adherence to COI disclosure and protocol registration was seen in randomized trials and reviews than in other research articles. The majority of research papers and reviews adhered to two or fewer indicators, whereas the majority of RCTs adhered to three or fewer indicators. Only 53 papers adhered to all indicators, overall. Journal- and publisher-related differences in transparency practices were clear. While some journals and publishers were completely transparent in terms of COI and funding disclosure, they had poor performance for the other three indicators. In addition, journals with higher JIF and H-index seemed to be more transparent, except for the COI disclosures. Otherwise, differences in transparent practices according to received citations and JIF were small.”

---

## [Decision Letter · Decision Letter 2]

26 May 2023

PONE-D-22-33929R2Transparency of COVID-19-related research: A meta-research studyPLOS ONE

Dear Dr. Sofi-Mahmudi,

Thank you for submitting your manuscript to PLOS ONE. After careful consideration, we feel that it has merit but does not fully meet PLOS ONE’s publication criteria as it currently stands. Therefore, we invite you to submit a revised version of the manuscript that addresses the points raised during the review process.

Dear authors,

thank you for sending the responses to the comments we raised and the corrections developed in the manuscript.

However, we have recently received the reviewer's comments for your manuscript in the context of statistics, focused on methods and reporting of the manuscript, which I am enclosing herewith.

I think it is important that you take into consideration and respond the questions and suggestions.

Major

1) The authors assume that all studies can share data. That is not the case. Many studies would use patient level registers and it would be gross misconduct in terms of the GDPR (which is not even mentioned in the paper, data security in general) if the data was available for everyone to download. So if the authors want to do this, they need to separate between studies that CAN and do not share data (population level, or aggregate freely available date) and analyses of practice level registries that cannot share data. Overall, this domain is deeply problematic and perhaps it should be discussed very carefully, if the authors want to keep it.

2) A second, not as problematic domain is the protocol registration. this is standard for RCTs and meta-analyses but not much else. of course it's good practice, but it is not expected in observational studies. so can the authors expand the introduction to provide more information on why this is needed and on any evidence on that leading to better work?

3) the data tool used is central to the work. The authors should provide performance metric for the tool, previously published (when the tool is described), in addition to their random sample.

4) I don't understand why the authors use piecemeal univariable analyses rather than use a regression model (linear or ordinal logistic) to examine factors associated with transparency, as they define it. the predictors can be JIF, year etc. this should overcome multiple testing issues and Type-I error inflation, if they use a single model. 

Minor

1) Report uncertainty in all the reported estimates (e.g. percentages in the abstract).

2) I found this to be unclear, please rephrase: "We used the interquartile range (the third quartile (Q3) – the first quartile (Q1)) when the data were skewed." Used for what purpose.

We look forward to receiving your revised manuscript.

Kind regards,

Rut Lucas-Dominguez, PhD

Academic Editor

PLOS ONE

Reviewers' comments:

Reviewer's Responses to Questions

**Comments to the Author**

1. If the authors have adequately addressed your comments raised in a previous round of review and you feel that this manuscript is now acceptable for publication, you may indicate that here to bypass the “Comments to the Author” section, enter your conflict of interest statement in the “Confidential to Editor” section, and submit your "Accept" recommendation.

Reviewer #4: (No Response)

2. Is the manuscript technically sound, and do the data support the conclusions?

Reviewer #4: Partly

3. Has the statistical analysis been performed appropriately and rigorously? 

Reviewer #4: No

4. Have the authors made all data underlying the findings in their manuscript fully available?

Reviewer #4: Yes

5. Is the manuscript presented in an intelligible fashion and written in standard English?

Reviewer #4: Yes

6. Review Comments to the Author

Reviewer #4: As the statistical reviewer I will focus on methods and reporting. The authors should note this is the first time I see the paper.

Major

1) The authors assume that all studies can share data. That is not the case. Many studies would use patient level registers and it would be gross misconduct in terms of the GDPR (which is not even mentioned in the paper, data security in general) if the data was available for everyone to download. So if the authors want to do this, they need to separate between studies that CAN and do not share data (population level, or aggregate freely available date) and analyses of practice level registries that cannot share data. Overall, this domain is deeply problematic and perhaps it should be discussed very carefully, if the authors want to keep it.

2) A second, not as problematic domain is the protocol registration. this is standard for RCTs and meta-analyses but not much else. of course it's good practice, but it is not expected in observational studies. so can the authors expand the introduction to provide more information on why this is needed and on any evidence on that leading to better work?

3) the data tool used is central to the work. The authors should provide performance metric for the tool, previously published (when the tool is described), in addition to their random sample.

4) I don't understand why the authors use piecemeal univariable analyses rather than use a regression model (linear or ordinal logistic) to examine factors associated with transparency, as they define it. the predictors can be JIF, year etc. this should overcome multiple testing issues and Type-I error inflation, if they use a single model.

Minor

1) Report uncertainty in all the reported estimates (e.g. percentages in the abstract).

2) I found this to be unclear, please rephrase: "We used the interquartile range (the third quartile (Q3) – the first quartile (Q1)) when the data were skewed." Used for what purpose.

7. PLOS authors have the option to publish the peer review history of their article (what does this mean?). If published, this will include your full peer review and any attached files.

Reviewer #4: No

---

## [Author Response · Author response to Decision Letter 2]

23 Jun 2023

Major

1) The authors assume that all studies can share data. That is not the case. Many studies would use patient level registers and it would be gross misconduct in terms of the GDPR (which is not even mentioned in the paper, data security in general) if the data was available for everyone to download. So if the authors want to do this, they need to separate between studies that CAN and do not share data (population level, or aggregate freely available date) and analyses of practice level registries that cannot share data. Overall, this domain is deeply problematic and perhaps it should be discussed very carefully, if the authors want to keep it.

2) A second, not as problematic domain is the protocol registration. this is standard for RCTs and meta-analyses but not much else. of course it's good practice, but it is not expected in observational studies. so can the authors expand the introduction to provide more information on why this is needed and on any evidence on that leading to better work?

Authors’ response: Thanks for your two important comments.

We agree and know that many papers may not have used data (such as commentaries, opinions, letters, etc.). To remove such papers from our analyses, we only considered the “research articles” filter of PubMed (access through EuropePMC). This filter should remove most papers that have not used any data.

We also agree that all the studies cannot publish their raw data. In such cases, those studies could have shared metadata. Metadata is descriptive information about the data's context, quality and condition, or characteristics. This perspective aligns with the FAIR principles we mentioned in the introduction. To clarify this in our paper, we have added an appropriate discussion of this distinction.(https://www.go-fair.org/fair-principles/).

As pointed out, the same challenges apply to code sharing and protocol registration. There have likely been studies for which either protocol registration, data or code sharing would have been unbeneficial or even impossible. We have expanded our introduction to provide more background on the importance of protocol registration and evidence that suggests it can lead to more rigorous research and also added this text (italics) to the discussion:

3rd paragraph:

Lacking protocol registration (for RCTs and reviews in particular), sharing code and data, and COI got attention during the pandemic (6,23,24)

While some research articles may not be able to share their data due to constraints such as privacy or legal obligations, at least most of them could have shared their metadata, that is “descriptive information about the context, quality and condition, or characteristics of the data” including “e.g., the data captured automatically by machines that generate data such as DICOM information for image files”.

5th paragraph:

First, with the applied methods we could not distinguish accurately all the studies that are required to register their protocol, e.g. distinguishing meta-analyses from narrative reviews; or studies that did not produce any (meta)data or code to share. So, this means that even though every researcher would have adhered to all five practices, we would not likely have achieved 100% adherence rates (e.g. because not every study uses any statistical procedures).

3) the data tool used is central to the work. The authors should provide performance metric for the tool, previously published (when the tool is described), in addition to their random sample.

Authors’ response: Thanks for the comment. Sensitivities and specificities of original validation by the Serghiou et al. were added to the method section as follows:

Sensitivity and specificity in the validation by the tool developers were in detection of

data sharing [sensitivity 0.76 (95% CI: 0.61-0.94), specificity 0.99 (95% CI: 0.98-1.00)]

code sharing [sensitivity 0.59 (95% CI: 0.34-0.94), specificity 1.00 (95% CI: 1.00-1.00)]

COI disclosures [sensitivity 0.99 (95% CI: 0.99-0.99), specificity 1.00 (95% CI: 0.99-1.00)]

funding disclosures [sensitivity 1.00 (95% CI: 0.99-1.00), specificity 0.98 (95% CI: 0.96-1.00)]

and protocol registration [sensitivity 0.96 (95% CI: 0.92-0.99), specificity 1.00 (95% CI: 1.00-1.00)].

Sensitivities and specificities of our validation sample of 100 articles are in the result-section and reads now as follows:

Of 100 random articles for validation, 20 discrepancies (out of 500 (five indicators per article), 4%) between the automatic tool and manual checking were found: 9% for open data [sensitivity 0.95 (95% CI: 0.89-0.99), specificity 0.62 (95% CI: 0.32-0.86)], 2% for open code [sensitivity 0.99 (95% CI: 0.94-1.00), specificity 0.67 (95% CI: 0.09-0.99)], 1% for COI disclosure [sensitivity 1.00 (95% CI: 0.72-1.00), specificity 0.99 (95% CI: 0.94-1.00)], 6% for funding disclosure [sensitivity 0.86 (95% CI: 0.68-0.96), specificity 0.97 (95% CI: 0.90-1.00)], and 2% for registration [sensitivity 0.99 (95% CI: 0.94-1.00), specificity 0.50 (95% CI: 0.01-0.99)] (S6 Appendix).

4) I don't understand why the authors use piecemeal univariable analyses rather than use a regression model (linear or ordinal logistic) to examine factors associated with transparency, as they define it. the predictors can be JIF, year etc. this should overcome multiple testing issues and Type-I error inflation, if they use a single model.

Authors’ response: Thanks for the comment. We agree and acknowledge that univariable analyses may not provide the depth of information that a regression model (linear or ordinal logistic) would afford. Therefore, we have followed your recommendation and implemented logistic regression analyses to scrutinize the association between articles published in journals To account for possible confounding factors, we adjusted these analyses for the year and month of publication and the Randomized Controlled Trial (RCT) status of the research article. These changes can be seen in the data analyses section, and the analysis is at the end of the results section. The discussion section was also edited accordingly.

Minor

1) Report uncertainty in all the reported estimates (e.g. percentages in the abstract).

Authors’ response: Done.

2) I found this to be unclear, please rephrase: "We used the interquartile range (the third quartile (Q3) – the first quartile (Q1)) when the data were skewed." Used for what purpose.

Authors’ response: Thanks for the comment. Now it reads:

We reported the interquartile range (the third quartile (Q3) – the first quartile (Q1)) and median (instead of mean and standard deviation) when the data were skewed.

---

## [Decision Letter · Decision Letter 3]

26 Jun 2023

Transparency of COVID-19-related research: A meta-research study

PONE-D-22-33929R3

Dear Dr. Ahmad Sofi-Mahmudi,

We’re pleased to inform you that your manuscript has been judged scientifically suitable for publication and will be formally accepted for publication once it meets all outstanding technical requirements.

Kind regards,

Rut Lucas-Dominguez, PhD

Academic Editor

PLOS ONE

Additional Editor Comments (optional):

All comments suggested by the reviewers have been addressed by the authors.

Reviewers' comments:

Reviewer's Responses to Questions

**Comments to the Author**

1. If the authors have adequately addressed your comments raised in a previous round of review and you feel that this manuscript is now acceptable for publication, you may indicate that here to bypass the “Comments to the Author” section, enter your conflict of interest statement in the “Confidential to Editor” section, and submit your "Accept" recommendation.

Reviewer #4: All comments have been addressed

2. Is the manuscript technically sound, and do the data support the conclusions?

Reviewer #4: Yes

3. Has the statistical analysis been performed appropriately and rigorously? 

Reviewer #4: Yes

4. Have the authors made all data underlying the findings in their manuscript fully available?

Reviewer #4: (No Response)

5. Is the manuscript presented in an intelligible fashion and written in standard English?

Reviewer #4: Yes

6. Review Comments to the Author

Reviewer #4: I am satisfied with the authors' responses.

I am satisfied with the authors' responses.

I am satisfied with the authors' responses.

7. PLOS authors have the option to publish the peer review history of their article (what does this mean?). If published, this will include your full peer review and any attached files.

Reviewer #4: No

---

## [Editor Report · Acceptance letter]

18 Jul 2023

PONE-D-22-33929R3 

Transparency of COVID-19-related research: A meta-research study 

Dear Dr. Sofi-Mahmudi:

I'm pleased to inform you that your manuscript has been deemed suitable for publication in PLOS ONE. Congratulations! Your manuscript is now with our production department. 

Kind regards, 

on behalf of

Prof. Rut Lucas-Dominguez 

Academic Editor

PLOS ONE